

# Adaptive robust observer-based control for structural load mitigation of wind turbines

Edwin Kipchirchir[1], M. Hung Do[2], Jackson G. Njiri[3], and Dirk Söffker[4]

[1,4]Chair of Dynamics and Control, University of Duisburg-Essen, Lotharstr. 1-21, 47057, Duisburg, Germany
[2]School of Heat Engineering and Refrigeration,Hanoi University of Science and Technology
[3]Department of Mechatronic Engineering, Jomo Kenyatta University of Agriculture and Technology, 62000-00200, Nairobi, Kenya

**Correspondence:** Edwin Kipchirchir (edwin.kipchirchir@uni-due.de)

**Abstract.** With growth in the physical size of wind turbines, an increased structural loading of wind turbine components affecting operational reliability is expected. To mitigate structural loading in wind turbines, a novel strategy for structural load mitigation and rotor speed regulation of utility-scale wind turbines in above-rated wind speed region is proposed in this contribution. Spatial and temporal variation of wind speed is responsible for fatigue loading during power production.

Previous attempts have proposed advanced control schemes incorporating disturbance models for cancelling the effects of wind disturbances. These controllers are usually designed based on reduced order models of nonlinear wind turbines, hence do not account for modeling errors and nonlinearities. Although robust controllers have been proposed to handle nonlinearities during wind turbine operation, these controllers are designed about specific operating points, hence suffer performance deterioration in changing operating conditions. In this contribution, a robust disturbance accommodating controller (RDAC), which is robust

against modeling errors and nonlinearities, is combined with an adaptive independent pitch controller (aIPC), designed to be adaptive to changing operating points due to wind speed variability, to mitigate structural loads in rotor blades and tower and to regulate rotor speed. The proposed control scheme is tested on a 1.5 MW National Renewable Energy Laboratory (NREL) reference wind turbine (RWT). Simulation results show that the proposed method successfully mitigates structural loading in rotor blades and tower without sacrificing rotor speed and power regulation performance in the presence of model uncertainties

and changing operating conditions.

## 1 Introduction

Renewables play a pivotal role in the transition to a net-zero energy system (Gielen et al., 2021). Wind energy, which has higher capacity factor compared to other renewables (Lee and Zhao, 2021), has grown over the past few decades from a niche to a mainstream source of renewable energy supported by improvements in wind turbine design and control technologies (Lee and

Zhao, 2020). The growth of wind turbine size over time has led to increased weight and flexibility of its components leading to increased structural loads during operation, which increases fatigue damage of wind turbine components. Finally, reliability due to downtime is affected, increasing Operation and Maintenance (O&M) costs known to contribute about 25 % to 40 % of the Levelized Cost Of Energy (LCOE) in wind turbines (Pfaffel et al., 2017).



In recent years, advanced wind turbine control strategies have been developed to mitigate structural loads, regulate rotor speed, and optimize power. However, most of these controllers are designed for specific operating points, and are not robust to modeling errors resulting from inherent wind turbine nonlinearities and operating point changes due to wind variability. In (Frost, 2009; Magar et al., 2013), adaptive collective pitch control (CPC) controllers are proposed to regulate rotor speed and accommodate wind disturbances. However, load mitigation is not considered. In (Corcuera et al., 2012), a $H_\infty$-based CPC controller is used for load reduction in drive-train and tower. Observer-based controllers, which are robust against modeling errors and nonlinearities, are proposed in (Do and Söffker, 2019, 2021) to mitigate tower loads using a CPC signals. However, these controllers are designed for given operating points.

While wind speed is stochastic in nature, vertical wind shear effect is deterministic. Vertical wind shear, tower shadow, and gravitational forces mainly affecting large wind turbines cause periodic 1P excitations of rotor blades and 3P excitations of the tower and fixed structure of the wind turbine (Kipchirchir et al., 2019). These periodic excitations can be reduced through independent pitch control (IPC). Disturbance accommodating control (DAC)-based IPC controllers are designed in (Wang et al., 2016, 2017) for mitigating 1P blade loads and regulating rotor speed. In (Njiri et al., 2019), a variable gain IPC scheme is proposed for wind turbine speed regulation and lifetime extension. However, only load mitigation of blades is considered and the controller is valid about a single operating point. In this contribution, the concepts proposed in (Do and Söffker, 2021) and (Njiri et al., 2019) are extended to develop an observer-based control strategy, which is not only robust to modeling errors but also adaptive to changes in operating point to regulate rotor speed and reduce structural loading in tower and rotor blades.

The paper is organized as follows. In section 2, the wind turbine model used in this work is described. In section 3, a brief overview of the robust observer-based controller for speed regulation is outlined. The proposed adaptive robust observer-based controller for rotor speed regulation and mitigation of tower and blades loads is described in section 4. In section 5, simulation results based on performance comparison of the proposed control scheme and RDAC controller on a reference wind turbine are discussed. Lastly, summary and conclusions are given in section 6.

## 2  Wind turbine model

A brief overview of the wind turbine model used for both the design and evaluation of the proposed control scheme is given.

Several RWTs reflecting current and future trends in the wind industry have been developed in recent years to study technologies for advancing performance of next generation of turbines (Rinker et al., 2020). A 1.5 MW WindPACT RWT developed by NREL (Rinker and Dykes, 2018), which is domicile in fatigue, aerodynamics, structures, and turbulence (FAST) design code (Jonkman and Buhl Jr., 2005), is chosen as the test-bed for the design and simulation of the proposed control strategy. This onshore wind turbine model whose specifications are summarized in Table 1, was developed based on a real-life commercial wind turbine used in the WindPACT program. It is a 3-bladed, upwind horizontal axis wind turbine, having 16 degrees of freedom (DoFs) describing its flexibility. However, a few DoFs are enabled to obtain reduced order LTI models used for controller design.



**Table 1.** 1.5 MW WindPACT reference wind turbine specifications

| Parameter | Value | Unit |
|---|---|---|
| Rated rotor speed | 20.463 | rpm |
| Hub height | 84.288 | m |
| Cut-in, Rated, Cut-out wind speed | 4, 12, 25 | m s$^{-1}$ |
| Gearbox ratio | 87.965 | - |
| Blade radius | 35 | m |
| Rated power | 1.5 | MW |
| Blade pitch range | 0-90 | $^o$ |
| Pitch rate | 10 | $^o$ s$^{-1}$ |
| Optimal Tip-Speed-Ratio ($\lambda_{opt}$) | 7.0 | - |
| Maximum power coefficient ($C_{pmax}$) | 0.5 | - |
| Optimum pitch angle ($\beta_{opt}$) | 2.6 | $^o$ |

The nonlinear generalized equation of motion for the wind turbine is expressed as

$$M(q,u,t)\ddot{q} + f(q,\dot{q},u,u_d,t) = 0, \qquad (1)$$

where $M$ denotes the mass matrix containing inertia and mass components, $f$ the nonlinear function of the enabled DoFs $q$ and their first derivative $\dot{q}$, as well as the control input $u$, disturbance input $u_d$, and time $t$. The nonlinear model Eq. (1) available

in FAST is linearized about an operating point in the above-rated wind speed region. By enabling the DoFs, which capture the most important wind turbine dynamics of interest, and specifying the operating point defined by a constant wind speed, blade pitch angle, and rotor speed, linearization is carried out numerically in FAST.

## 3   Robust observer-based control

A RDAC controller for rotor speed regulation and tower load mitigation, is introduced. This controller proposed in previous

work (Do and Söffker, 2021), is extended with the view of meeting additional objectives. It is briefly repeated here for principal understanding.

To obtain a linear model for controller design, the nonlinear model Eq. (1) is linearized about an operating point in the high wind speed regime defined by a constant hub-height wind speed of $v_{op}$=18 m s$^{-1}$, a pitch angle of $\beta_{op} = 20^o$, and a rotor speed of $\omega_{op} = 20.463$ rpm. To capture the most important dynamics, corresponding to the desired closed-loop performance with

70 respect to structural load mitigation and rotor speed regulation, the mechanical states $x \in \mathbb{R}^{11x1}$ utilized for controller design





are

$$x = \begin{bmatrix} \text{tower-top fore-aft displacement} \\ \text{drivetrain torsional displacement} \\ \text{blade 1 flap-wise displacement} \\ \text{blade 2 flap-wise displacement} \\ \text{blade 3 flap-wise displacement} \\ \text{generator speed} \\ \text{tower fore-aft velocity} \\ \text{drivetrain torsional velocity} \\ \text{blade 1 flap-wise velocity} \\ \text{blade 2 flap-wise velocity} \\ \text{blade 3 flap-wise velocity} \end{bmatrix}. \tag{2}$$

The obtained reduced-order LTI model is expressed in state-space form as

$$\dot{x} = Ax + Bu + B_d d$$
$$y = Cx, \tag{3}$$

where $A \in \mathbb{R}^{11x11}$ denotes the linearized system matrix, $B \in \mathbb{R}^{11x1}$ the control input matrix, $B_d \in \mathbb{R}^{11x1}$ the disturbance matrix, and $C \in \mathbb{R}^{2x11}$ denotes the output matrix, $u \in \mathbb{R}^{1x1}$ the perturbed collective pitch angle $\Delta\beta$, $d \in \mathbb{R}^{1x1}$ the perturbed hub-height wind speed $\Delta v$. The measurements $y \in \mathbb{R}^{2x1}$ include rotor speed $\omega$ and tower-base fore-aft bending moment.

To account for pitch actuator dynamics, which is not integrated in FAST, Eq. (3) is extended to include a pitch actuator model. Because pitch actuator dynamics are faster than other wind turbine dynamics, it is modeled as a first-order lag (PT1)

linear model

$$\frac{\beta}{\beta_{com}} = \frac{1}{s\tau_\beta + 1}, \tag{4}$$

where $\beta_{com}$ denotes the commanded pitch angle, $\beta$ the actual pitch angle, and $\tau_\beta$ the actuator time constant. The model is represented in state-space form as

$$\dot{\beta} = -\frac{1}{\tau_\beta}\beta + \frac{1}{\tau_\beta}\beta_{com}. \tag{5}$$

Using Eq. (5), model (3) is extended to obtain the model

$$\underbrace{\begin{bmatrix} \dot{x} \\ \dot{\beta} \end{bmatrix}}_{\dot{x}_a} = \underbrace{\begin{bmatrix} A & B \\ 0 & -1/\tau_\beta \end{bmatrix}}_{A_a} \underbrace{\begin{bmatrix} x \\ \beta \end{bmatrix}}_{x_a} + \underbrace{\begin{bmatrix} 0 \\ 1/\tau_\beta \end{bmatrix}}_{B_a} u + \underbrace{\begin{bmatrix} B_d \\ 0 \end{bmatrix}}_{B_{da}} d$$

$$y = \underbrace{\begin{bmatrix} C & 0 \end{bmatrix}}_{C_a} \begin{bmatrix} x \\ \beta \end{bmatrix}. \tag{6}$$



### 3.1 Disturbance accommodating control for wind turbines

Highly turbulent wind conditions influence the power and torque of a wind turbine as well as causing cyclic loading of its components. Therefore, there is need to counteract these wind disturbances without affecting full-state feedback and observability.

Assuming the disturbance structure is known, wind disturbance states can be added to the extended model Eq. (6) to design a DAC controller. This controller estimates disturbance states by augmenting the observer-based controller with an assumed waveform model (Wright, 2004). The special disturbance observer gain is then used to cancel disturbance effects.

In the realm of wind turbines, spatial variation of rotor effective wind speed is considered an additive disturbance having a waveform model of the form

$$
\begin{aligned}
d &= \theta x_d \\
\dot{x}_d &= F x_d,
\end{aligned}
\tag{7}
$$

where $x_d$ denotes the wind disturbance state while $\theta$ and $F$ denote the known disturbance state space model. Assuming a step disturbance waveform, which approximates sudden uniform rotor effective wind velocity fluctuations, the state-space matrices are chosen as $\theta = 1$ and $F = 0$ (Söffker et al., 1995; Wright, 2004; Wright and Fingersh, 2008). Model Eq. (6) is extended to include the wind disturbance model as

$$
\underbrace{\begin{bmatrix} \dot{x}_a \\ \dot{x}_d \end{bmatrix}}_{\dot{x}_e} = \underbrace{\begin{bmatrix} A_a & B_{da}\theta \\ 0 & F \end{bmatrix}}_{A_e} \underbrace{\begin{bmatrix} x_a \\ x_d \end{bmatrix}}_{x_e} + \underbrace{\begin{bmatrix} B_a \\ 0 \end{bmatrix}}_{B_e} u
$$

$$
y = \underbrace{\begin{bmatrix} C_a & 0 \end{bmatrix}}_{C_e} \begin{bmatrix} x_a \\ x_d \end{bmatrix}.
\tag{8}
$$

After establishing full observability, system and disturbance states are estimated by designing the extended observer

$$
\begin{bmatrix} \dot{\hat{x}}_a \\ \dot{\hat{x}}_d \end{bmatrix} = \begin{bmatrix} A_a & B_{da}\theta \\ 0 & F \end{bmatrix} \begin{bmatrix} \hat{x}_a \\ \hat{x}_d \end{bmatrix} + \begin{bmatrix} B_a \\ 0 \end{bmatrix} u + L(y - \hat{y})
$$

$$
\hat{y} = \begin{bmatrix} C_a & 0 \end{bmatrix} \begin{bmatrix} \hat{x}_a \\ \hat{x}_d \end{bmatrix},
\tag{9}
$$

where the observer gain $L$ is typically calculated using pole placement or LQR method. Using the estimated states, full-state feedback control is implemented as

$$
u = u_x + u_d = -K_x \hat{x}_a - K_d \hat{x}_d,
\tag{10}
$$

where $K_x$ denotes the full-state feedback controller used to realize speed regulation and structural load mitigation control objectives and $K_d$ denotes the disturbance rejection controller designed separately to cancel wind disturbances effects. Using





the control variable in Eq. (10), Eq. (9) can be rewritten as

$$
\begin{bmatrix} \dot{\hat{x}}_a \\ \dot{\hat{x}}_d \end{bmatrix} = \begin{bmatrix} A_a & B_{da}\theta \\ 0 & F \end{bmatrix} \begin{bmatrix} \hat{x}_a \\ \hat{x}_d \end{bmatrix} + \begin{bmatrix} B_a \\ 0 \end{bmatrix} \begin{bmatrix} K_x & K_d \end{bmatrix} \begin{bmatrix} \hat{x}_a \\ \hat{x}_d \end{bmatrix} - \underbrace{\begin{bmatrix} L_1 \\ L_2 \end{bmatrix}}_{L} \begin{bmatrix} C_a & 0 \end{bmatrix} \begin{bmatrix} \hat{x}_a \\ \hat{x}_d \end{bmatrix} + Ly,
\tag{11}
$$

where $L_1$ denotes the system observer gain matrix and $L_2$ the disturbance observer gain matrix. The observer gains are typically calculated using pole placement (Söffker et al., 1995), or LQR method. While $K_x$ is usually designed using pole placement or LQR technique, in standard DAC approaches, $K_d$ is chosen to minimize the norm $\|B_a K_d + B_{da}\theta\|$ by using Moore-Penrose Pseudoinverse (†) or Kronecker Product method.

   To meet the objective of rotor speed regulation with zero static tracking error, the DAC model Eq. (11) is extended with a

partial integral action $\dot{x}_i = C_i y$, where $C_i$ denotes the location of the measured rotor speed in the output while $K_i$ denotes the integral gain. Therefore, the dynamic DAC controller with partial integral action becomes

$$
\underbrace{\begin{bmatrix} \dot{\hat{x}}_a \\ \dot{\hat{x}}_d \\ \dot{x}_i \end{bmatrix}}_{\dot{x}_r} = \underbrace{\begin{bmatrix} A_a - B_a K_x - L_1 C_a & B_{da}\theta + B_a K_d & B_a K_i \\ -L_2 C_a & F & 0 \\ 0 & 0 & 0 \end{bmatrix}}_{A_r} \underbrace{\begin{bmatrix} \hat{x}_a \\ \hat{x}_d \\ x_i \end{bmatrix}}_{x_r} + \underbrace{\begin{bmatrix} L_1 \\ L_2 \\ C_i \end{bmatrix}}_{B_r} y,
$$

$$
u = \underbrace{\begin{bmatrix} K_x & K_d & K_i \end{bmatrix}}_{C_r} \begin{bmatrix} \hat{x}_a \\ \hat{x}_d \\ x_i \end{bmatrix}.
\tag{12}
$$

   In existing approaches, DAC parameters including observer gain $L$, and state controller $K_x$ and disturbance rejection controller $K_d$ are calculated separately, without considering the closed loop system stability, robustness and optimality. Therefore,

a robust control method for obtaining optimal DAC parameters in a single step is required.

### 3.2   Robust disturbance accommodating control

The standard H$_\infty$ control problem is usually formulated as a task to minimize the H$_\infty$ norm $\|.\|_\infty$ of the transfer function $G_{zd}$ from the exogenous inputs $d$ to the controlled outputs $z$ as

$$
R^* = \underset{R \in \mathcal{R}}{argmin} \| G_{zd}(P,R) \|_\infty,
\tag{13}
$$

where $R^*$ denotes the optimized controller, $\mathcal{R}$ a set of controllers $R$ that stabilize the plant $P$. The effects of exogenous disturbances on the outputs is minimized by using $R^*$, hence increasing system robustness. This convex optimization problem can be solved using algebraic Riccati equations (ARE) or linear matrix inequalities (LMI). Standard H$_\infty$ control can not be applied in control systems with structural constraints like the structured DAC control system Eq. (12), which depends smoothly on the design parameters $L$, $K_x$, $K_d$, and $K_i$.





To provide a trade-off between robust stability and performance, weighting functions are usually introduced. Therefore, the optimization problem Eq. (13) is extended to become a mixed-sensitivity H$_\infty$ problem expressed as

$$R^* = \underset{R \in \mathcal{R}}{argmin} \left\| \begin{matrix} W_1 S \\ W_2 RS \\ W_3 T \end{matrix} \right\|_\infty, \tag{14}$$

where, $W_1$, $W_2$, and $W_3$ denote the weighting functions, while $S$, $RS$, and $T$ denote the sensitivity function, control effort, and complementary sensitivity function, respectively. This serves as a cost function for optimizing parameters of a structured DAC controller. The problem to find the optimal RDAC controller $RDAC^*$ defining the optimal parameters $K^* = [K_x \ K_d \ K_i]$ and $L^* = [L_1 \ L_2]^T$ is formulated as

$$RDAC^* = \underset{RDAC \in \mathcal{RDAC}}{argmin} \| G_{zd}(P, RDAC) \|_\infty, \tag{15}$$

where $\mathcal{RDAC}$ denotes a set of controllers $RDAC$ that stabilize the generalized plant $P$ made up of weighting functions, pitch actuator dynamics, and wind turbine nominal model.

To ensure that $RDAC^*$ guarantees asymptotic stability of the closed loop system, the optimization problem Eq. (15) is subjected to the Lyapunov stability constraint $\| C_a(sI - \mathcal{A}(RDAC))^{-1} B_a \|_\infty < +\infty$, where $\mathcal{A}(RDAC)$ denotes the closed-loop system matrix depending on RDAC controller. Therefore, the optimization problem becomes

$$RDAC = \underset{RDAC \in \mathcal{RDAC}}{argmin} \| G_{zd}(P, RDAC) \|_\infty$$
$$s.t. \ \| C_a(sI - \mathcal{A}(RDAC))^{-1} B_a \|_\infty < +\infty, \tag{16}$$

whose H$_\infty$ norms are calculated from the closed-loop system using a bisection algorithm. Since Eq. (16) is non-convex, hence cannot be solved using AREs or LMIs, nonsmooth H$_\infty$ synthesis (Apkarian and Noll, 2006), used for problems with structural and stability constraints is applied to find an optimal controller $RDAC^*$ for tower load mitigation and rotor speed regulation of wind turbines. This method is implemented in MATLAB using *hinfStruct* command (Apkarian and Noll, 2017). It uses Clarke sub-differential and a multi-start steepest gradient descent method to minimize the H$_\infty$ norms.

The proposed RDAC approach is applied to the 1.5 MW NREL reference wind turbine (Fig. 1). To account for blade pitch actuator dynamics, an actuator transfer function is included in the generalized plant P. Hub-height wind disturbance d excites the wind turbine dynamics in the above-rated wind speed region. The RDAC controller relies on measured outputs, which includes rotor speed $\omega$ and tower fore-aft bending moment $\zeta$ to generate a collective pitch angle $\beta$ as a control signal for regulating rotor speed at the rated value and reducing tower fore-aft bending moment oscillations. The RDAC controller is designed to be robust against modeling errors and wind disturbances. The weighting functions W$_{11}$, W$_{12}$, and W$_2$ are designed to obtain desired robust performance. To effect rotor speed response and ensure robustness against wind disturbances W$_{11}$ is designed as an inverted low-pass filter. To reduce the first mode of tower fore-aft oscillation, which occurs at 2.56 rad/s, W$_{12}$ is designed as an inverted notch filter centered at this frequency. Finally, to reduce controller activity at high frequencies thereby increasing robustness, W$_2$ is chosen as an inverted high-pass filter.



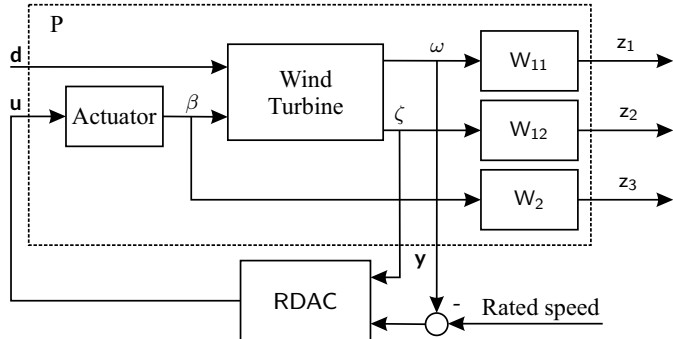

**Figure 1.** RDAC for wind turbines.

The RDAC controller, which is robust against modeling errors and wind disturbances ensures both objectives of rotor speed
regulation and tower load reduction for wind turbines in above-rated operation are met. However, the controller is only valid
within its design operating point and suffers performance deterioration outside this envelop. Additionally, its control input
signal is a collective pitch angle, hence cannot be applied for reducing blade oscillations due to vertical wind shear as this can
only be achieved through IPC.

## 4 Modified robust disturbance accommodating control using adaptive independent pitch control

A novel adaptive independent pitch controller (aIPC) designed using LQG method and implemented as a separate control
loop is proposed to enhance the performance of the RDAC controller. It is designed to reduce 1P (0.333 Hz) blade flap-wise
oscillations and is adaptive to change in operating point due to horizontal wind speed fluctuations.

### 4.1 Adaptive independent pitch control

As wind turbine rotor blades rotate, they experience varying aerodynamic loads at different azimuth positions due to vertical
wind shear. To counteract this periodicity, which is more pronounced in large wind turbines, rotor blades are pitched inde-
pendently. The idea behind aIPC is using 5 IPC controllers, each designed to be effective over a specified wind speed bin in
above-rated operation. The linear models used for designing respective IPC controllers, are extracted from the nonlinear wind
turbine model Eq. (1) at different operating points shown in Table 2.

To capture the most important dynamics with respect to blade load mitigation and rotor speed regulation, 7 states $x$ includ-
ing blade flap-wise displacement for each blade and their respective velocities, and generator speed are selected. To capture
periodicity in aerodynamic loading due to vertical wind shear, 24 equispaced azimuth positions are selected for linearization.
To integrate this inherent periodicity in the controller design, multi-blade coordinate (MBC) transformation (Bir, 2010) is used
to transform the individual blade dynamics in the rotating frame to the non-rotating frame. The transformed reduced order





**Table 2.** Design operating points for the IPC controllers

| IPC Controller | Wind speed bin [m s$^{-1}$] | Steady wind speed [m s$^{-1}$] | Blade pitch angle [$^o$] | Rotor speed [rpm] |
|---|---|---|---|---|
| 1 | 12 - 15 | 14 | 13.10 | 20 |
| 2 | 15 - 17 | 16 | 16.75 | 20 |
| 3 | 17 - 19 | 18 | 19.83 | 20 |
| 4 | 19 - 21 | 20 | 22.47 | 20 |
| 5 | 21 - 25 | 22 | 24.84 | 20 |

models are then averaged to obtain a weakly periodic LTI model described in state-space form as

$$\dot{x} = Ax + Bu + B_d d$$
$$y = Cx, \tag{17}$$

where $u \in \mathbb{R}^{3x1} = [\Delta\beta_1 \ \Delta\beta_2 \ \Delta\beta_3]^T$ denotes the perturbed independent pitch angles, $d \in \mathbb{R}^{1x1}$ the wind disturbance. The matrices $A \in \mathbb{R}^{7x7}$, $B \in \mathbb{R}^{7x1}$, $B_d \in \mathbb{R}^{7x1}$, and $C \in \mathbb{R}^{3x7}$ denote the system, input, disturbance, and output matrices, respectively. The measurements $y \in \mathbb{R}^{3x1}$, which include the blade root flap-wise bending moment for each blade, are assumed to be distorted with noise $v$.

To implement full-state feedback control, the control gain matrix $K = R^{-1}B^T P$ is designed using LQR technique by minimizing the quadratic performance index

$$J_{QR} = \int\limits_0^\infty (x^T Q x + u^T R u)dt, \tag{18}$$

while solving the ARE $A^T P + PA - PBR^{-1}B^T P + Q = 0$ assuming $(A, B)$ is fully controllable. Here, $Q$ and $R$ denote symmetric positive definite state and control input weighting matrices respectively, whose elements are chosen to achieve desired dynamic response with respect to blade load mitigation and rotor speed regulation. The symmetric positive definite matrix $P$ is the solution to the ARE.

Some states might not be available for measurement and additionally, it is cost-effective to use a few measurements to reconstruct systems states. Given that wind turbine dynamics are excited by stochastic wind profiles and that measurement signals are typically noisy, a Kalman state estimator is used to obtain estimated states $\hat{x}$ for implementing full-state feedback control. The process noise $d$ and measurement noise $v$ are assumed to be uncorrelated zero mean Gaussian white noise with process disturbance covariance matrix $Q_f = E(ww^T)$ and measurement noise covariance matrix $R_f = E(vv^T)$.

After determining that $(A, C)$ is fully observable, the observer gain $L = P_f C^T R^{-1}$ is designed by minimizing the state estimation covariance error $E((x - \hat{x})(x - \hat{x})^T)$, while solving the filter algebraic Riccati equation (FARE) $AP_f + P_f A^T - P_f C^T R_f^{-1} C P_f + Q_f = 0$, where $P_f = P^T \geq 0$ is the solution to the FARE. An optimal full-state feedback control is implemented using the estimated states as $u = -K\hat{x}$.





An implementation of one of the five IPC controllers is shown in Fig. 2. The wind profile $d$ excites the dynamics of the wind turbine in above-rated regime. The perturbed blade root flap-wise bending moment measurements $\Delta y$ are transformed from the rotating to the fixed coordinate frame of controller design, using the inverse MBC transformation matrix $T(\psi)^{-1}$, which relies on real-time rotor azimuth angle measurements $\psi$. The perturbed independent pitch angles $\Delta\beta_i$ are obtained by transforming

the control input $u$ back to the rotating coordinate frame using the MBC transformation matrix $T(\psi)$. By summing $\Delta\beta_i$ and the collective pitch angle $\beta_c$ from the RDAC controller, the IPC signal $\beta_i$ obtained. A technique for switching between the IPC controllers based on the incoming wind speed is implemented in MATLAB/Simulink. This constitutes the aIPC controller.

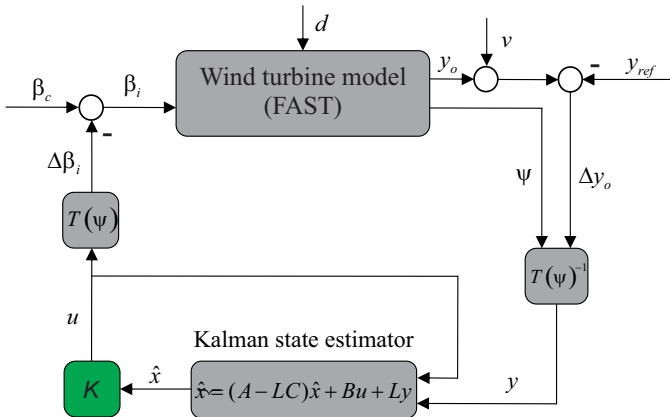

**Figure 2.** Independent pitch controller

## 4.2 Adaptive robust observer-based control

As depicted in Figure 3, adaptive robust observer-based controller (RDAC+aIPC) is implemented using two control loops. For

above-rated operation, generator torque is kept constant at the rated value. The RDAC controller relies on tower-base fore-aft bending moment and generator speed measurements to generate a collective pitch angle signal for tower load mitigation and rotor speed regulation. The aIPC controller relies on blade-root flap-wise bending moments and azimuth measurements to generate IPC signals used for blade load mitigation. The independent pitch angles are perturbed about the CPC signal forming the control input for the wind turbine model in FAST. Although reduced order models are used for designing the two

controllers, the full-order 1.5 MW NREL RWT is used for simulation.





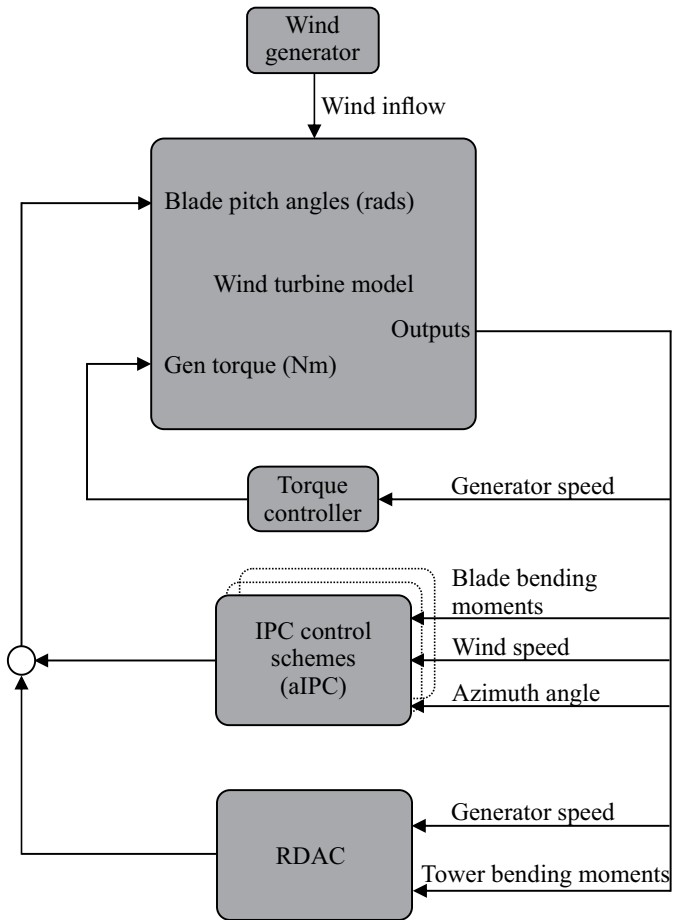

**Figure 3.** Adaptive robust observer-based controller

# 5   Results and discussion

Simulation results obtained from evaluating the adaptive robust observer-based control strategy using the 1.5 MW NREL RWT in FAST design code are discussed. Both step and stochastic wind profiles are used to excite the wind turbine dynamics in the above-rated wind speed region. Performance of the proposed control schemes are compared. Selected performance measures

are used for comparison.

## 5.1   Performance measures for analyzing results

Time-series results obtained from simulation are analyzed in time and frequency domain using a number of criteria to illustrate improvements in structural load mitigation and speed/power regulation of the proposed control scheme.





### 5.1.1 Time domain

The time-series results analyzed include blade flap-wise and tower fore-aft bending moments, rotor rotational speed, generator power, and blade pitch angles. The time-series values are plotted for a graphical illustration of the results. Additionally, mean and standard deviation of the data is computed.

### 5.1.2 Frequency domain

While time domain analysis gives an incite into the temporal behaviour of signals, spectral analysis is necessary for estimating 230 the frequency components in a time-series signal. Power spectral density (PSD) analysis using Welch's method (Welch, 1967) is used to obtain the frequency-dependence of the time-series structural loading data from simulation. To improve the spectral estimation process, the method uses a windowing mechanism to shape the time-series signal before its PSD is computed. To achieve this, time-series signal is divided into little time-slices called windows, with the percentage of overlap between windows being specified. Average of the fast Fourier transform (FFT) of all windows is then performed to obtain a smooth 235 signal, whose PSD is then computed.

In this work, spectral analysis is used to analyze the contribution of 1P and 3P frequencies in blade and tower loading.

### 5.1.3 Power-load covariance

While previously outlined performance measures can yield meaningful information on structural load mitigation or rotor speed regulation performance, they do not consider the relationship between structural loads and speed/power regulation.

To get a clear illustration of the performance of the proposed control scheme with respect to both load mitigation and power regulation, power-load covariance criteria proposed in (Do and Soeffker, 2020) is used. In this graphical method a power-load distribution diagram and ellipse iso-contours based on the power and load covariance matrices are used to obtain five performance measures. These include average and variance in both power and structural load, and the power-load covariance level.

### 245 5.2 Step wind profile results

To evaluate the closed-loop control performance under changing operating point, a step wind profile shown in Fig. 4a is used. It has vertical wind shear with a conservative power-law exponent of 0.2. The hub-height wind speed varies in steps from 14 m/s to 22 m/s. To compare the pitch activity of different blades, pitch angles of the CPC signal from RDAC and the IPC signals from RDAC+aIPC are plotted as illustrated in Fig. 4b. Additional pitching of each blade about the CPC signal provided by 250 the IPC controllers active at different operating points to mitigate cyclic loading of blades due to vertical wind shear occurs. Additionally, smooth switching between the different controllers is realized.

Structural load mitigation performance in rotor blades and tower is also evaluated. As illustrated in Fig. 5a, the RDAC+aIPC controller shows a significant reduction in the blade-root flap-wise bending moment vibration amplitude. The standard devi-



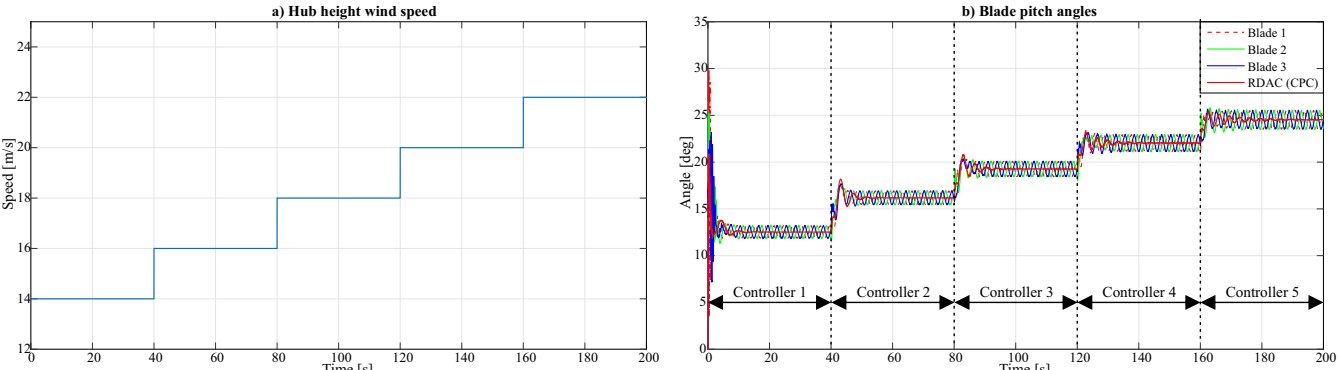

**Figure 4.** Step wind response

ation reduces by 13.7 % compared with the RDAC controller. Mitigation of tower-base fore-aft bending moment follows a similar trend as shown in Fig. 5b, with 8 % reduction in standard deviation.

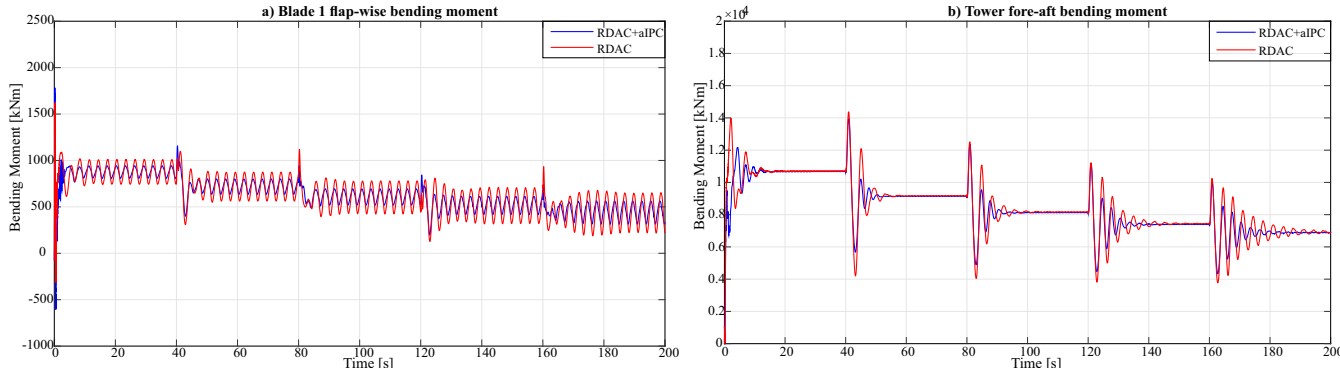

**Figure 5.** Structural loading in blades and tower

Despite the improved performance in structural load mitigation achieved by the proposed controller, there is need to ascertain that this does not come at a trade-off in speed/power regulation performance. Rotor speed measurement is used to evaluate speed regulation as shown in Fig. 6a. The proposed controller shows improved transient performance attributed to aIPC, with a 1.23 % reduction in standard deviation. However, it shows slightly reduced power regulation performance Fig. 6b, which is

260 attributed to high pitching especially in high wind speeds. The standard deviation in generator power increases by 5.73 %.

Spectral analysis of the time-series blade loading shows a significant reduction in 1P and around 3P frequency components of blade flap-wise bending moment as illustrated in Fig. 7a. There is also reduction in both 1P and 3P frequency components of tower fore-aft bending moment as depicted in Fig. 7b. This is attributed to aIPC, which reduces 1P vibration amplitude of each blade, leading to reduced 3P tower oscillation.

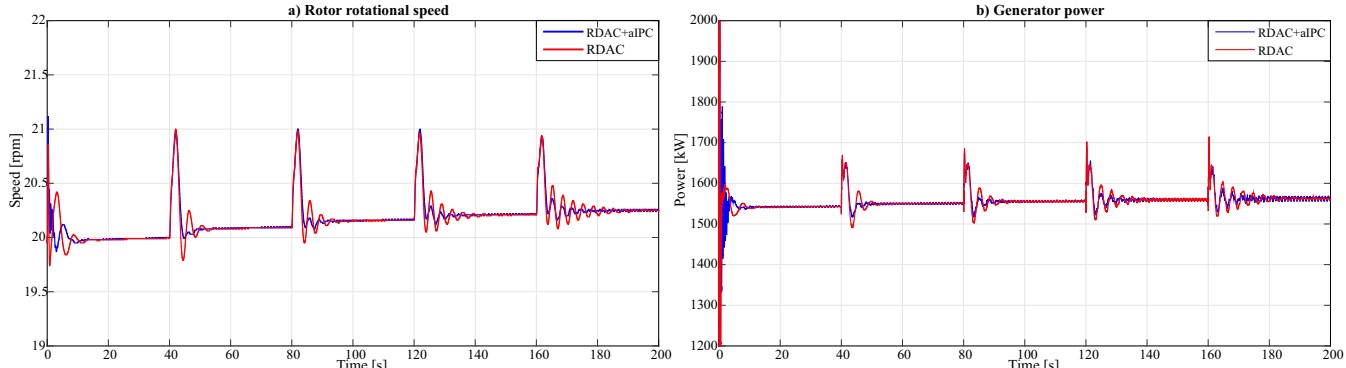

**Figure 6.** Speed/power regulation response

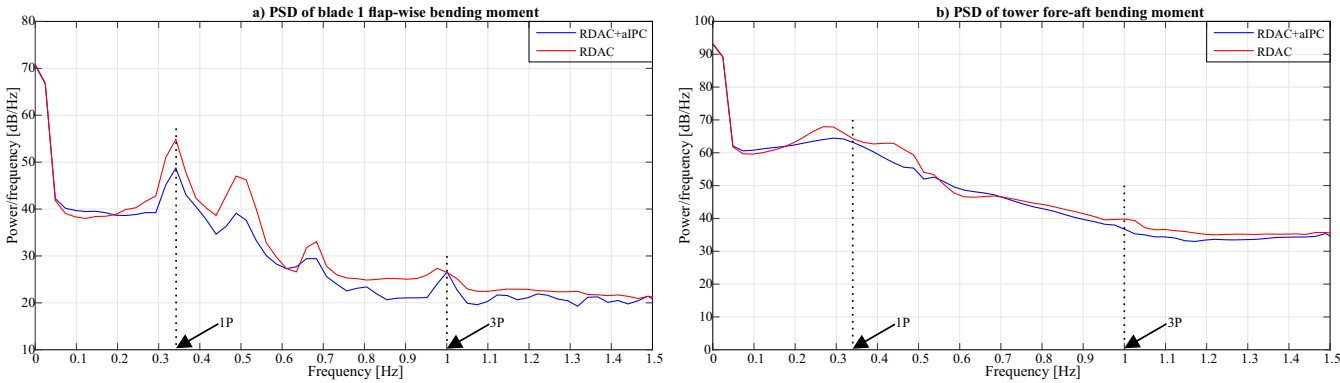

**Figure 7.** Spectral analysis of blade and tower loading

To give a clear illustration of the control performance in both load mitigation and speed/power regulation, generator power versus structural load covariance is evaluated (Fig. 8). The proposed control strategy shows lower variance in both blade and tower loads $\sigma_{x2}$, as well as in the generated power $\sigma_{y2}$. Therefore, there is improved structural load mitigation without trade-off in speed/power regulation performance.

### 5.3 Stochastic Wind Profile Results

To evaluate the closed-loop performance of the proposed control strategy under more realistic wind conditions, a stochastic wind profile shown in Fig. 9a is used. The IEC von Karman type A wind profile generated using TurbSim software (Jonkman and Kilcher, 2012) has a turbulence intensity 16 % at 15 m/s, and a mean hub-height wind speed is 18 m/s. It has a vertical wind shear with a power law exponent of 0.2. As depicted in Fig. 9b, there is additional pitching of each blade by the aIPC control signal about the CPC signal to mitigate vertical wind shear effect.



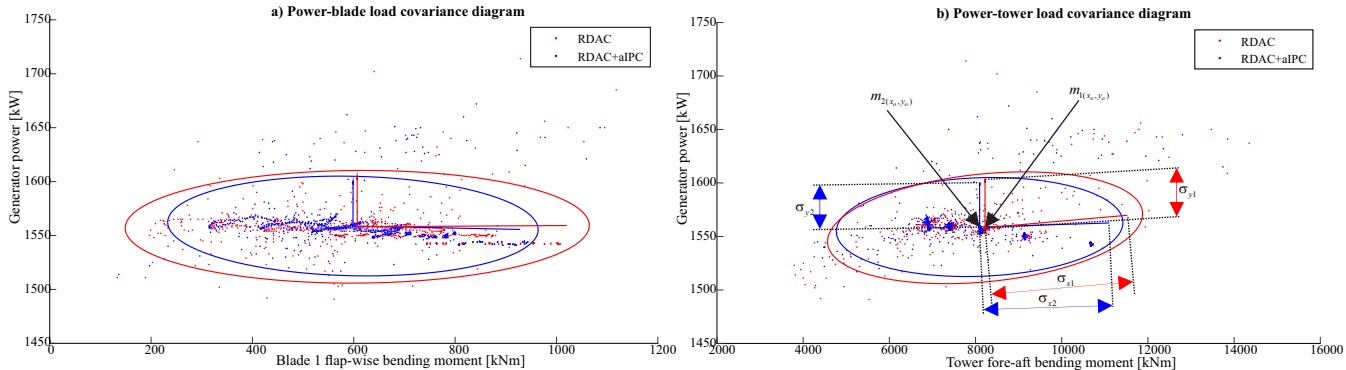

**Figure 8.** Power-load covariance analysis

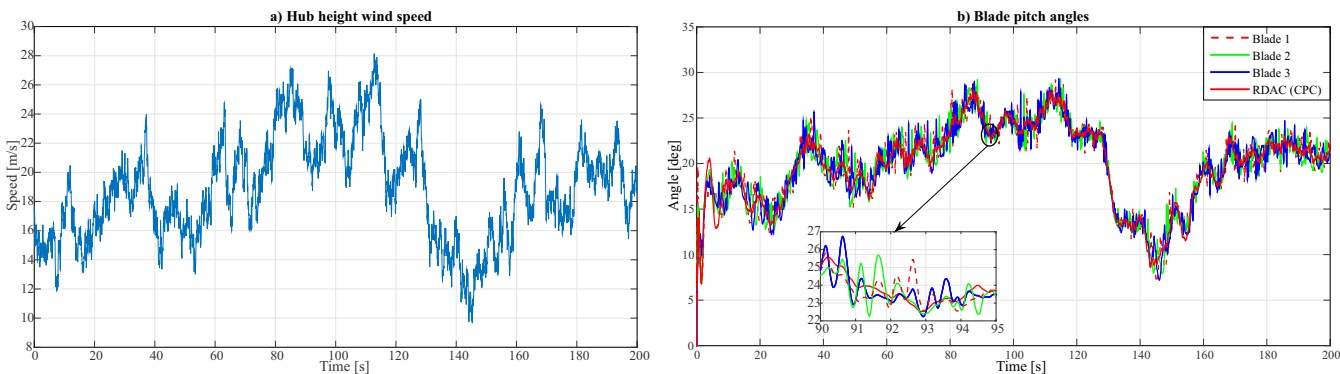

**Figure 9.** Stochastic wind response

Structural load mitigation performance in the blades and tower is illustrated in Fig. 10. As compared with RDAC, the proposed control scheme shows a significant reduction in blade flap-wise and tower fore-aft bending moment variations, with the standard deviations reducing by 11.9 % and 12.04 % respectively. However, the mean loading values remain almost unchanged.

The speed/power regulation performance of the proposed control scheme as compared to RDAC controller is illustrated in Fig. 11. It can be seen that a slight deterioration in both rotor speed and power regulation performance occurs, with the standard deviation increasing by 3.14 % and 17.7 % respectively. This is attributed to high pitching activity in high wind speeds. Additionally, highly turbulent wind dynamics is faster than the pitch dynamics. However, the mean value of generated power remains unchanged.

Spectral analysis of the blade-flap-wise bending moment depicted in Fig. 12a shows reduction in 3P frequency component with little dependency at 1P frequency. The PSD of the time-series tower fore-aft bending moment follows a similar trend as can be seen Fig. 12b. Therefore, decreased asymmetrical loading of the blades contributes to reduced tower vibration at 3P frequency.




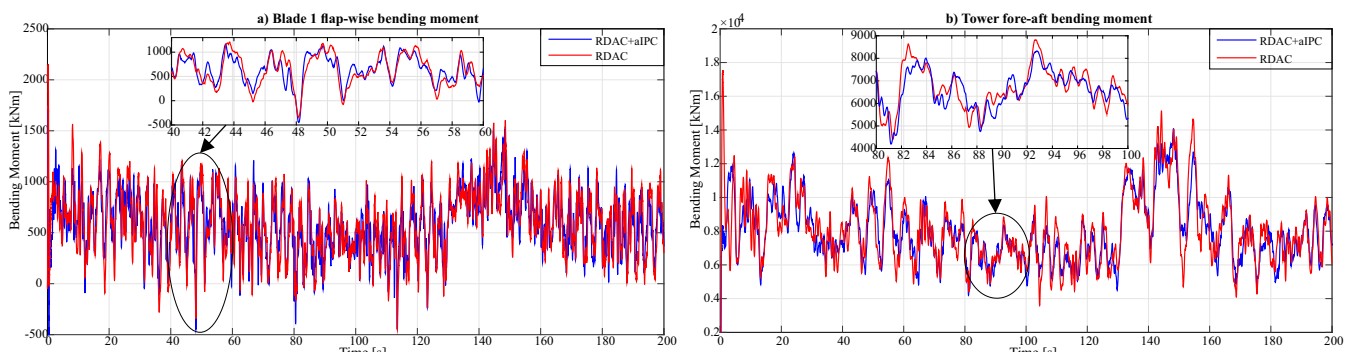

**Figure 10.** Structural loading in blades and tower

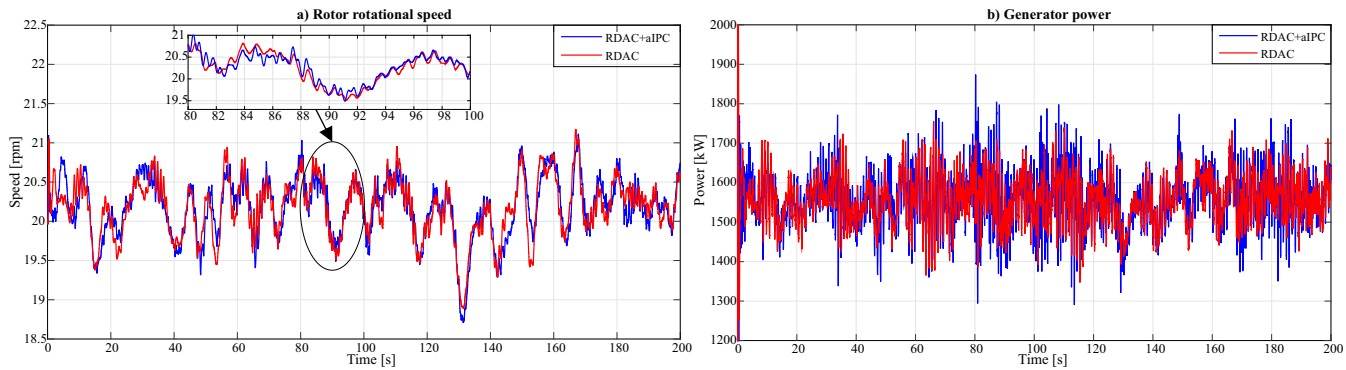

**Figure 11.** Speed/power regulation response

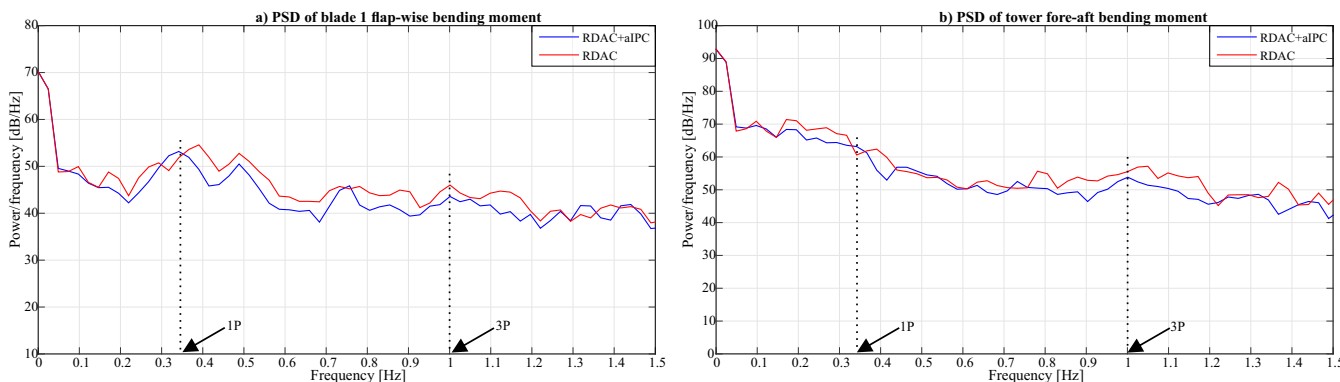

**Figure 12.** Spectral analysis of blade and tower loading

Generator power versus structural load covariance evaluation illustrated in Fig. 13 shows that the proposed control strategy exhibits lower variance in both blade and tower loads albeit with a slightly increased variance in the generated power. Therefore, the adaptive robust observer-based controller improves structural load mitigation without significant power regulation trade-off.



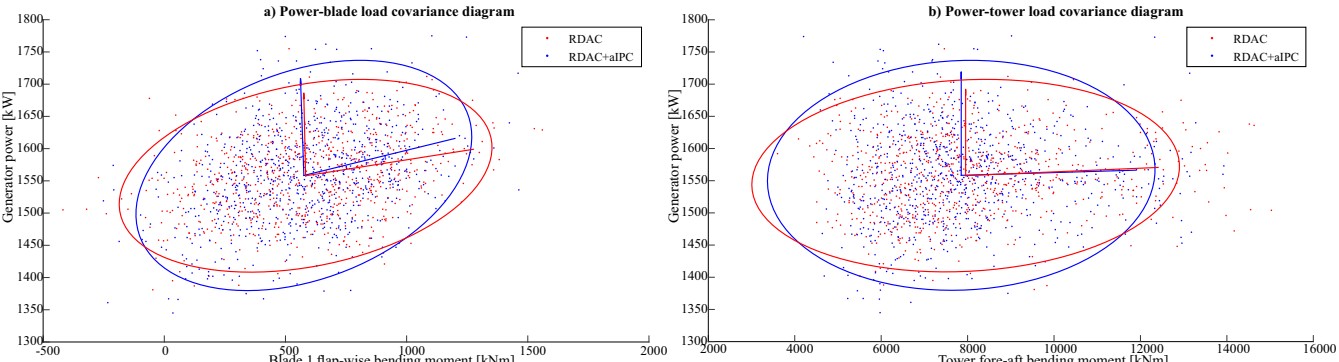

**Figure 13.** Power load covariance analysis

## 6 Summary and Conclussion

In this contribution, a robust disturbance accommodating controller (RDAC) extended with a novel adaptive independent pitch controller (aIPC) is presented. This adaptive robust observer-based control scheme is used to mitigate structural loading in both the tower and blades and to regulate rotor speed for wind turbines operating in the above-rated wind speed region. The RDAC controller used for tower load mitigation and speed regulation, is designed by minimizing the mixed sensitivity $H_\infty$ norm of the generalized wind turbine system using nonsmooth $H_\infty$ synthesis. On the other hand, the aIPC is designed using LQG method for blade load mitigation. Simulation results obtained under step wind conditions shows that the proposed control scheme provides better performance in both structural load reduction and speed regulation as compared to the RDAC controller. Although slight deterioration in rotor speed regulation is seen under stochastic wind conditions, improvement in structural load mitigation far outweighs this setback. The proposed method is robust to wind turbine modeling errors and nonlinearities, and is adaptive to operating point changes due to wind disturbances.

*Code availability.* Code is not publicly available and can not be shared.

*Author contributions.* DS and MHD proposed the original idea of using robust control method for optimizing the DAC gain matrices to obtain a RDAC controller, which was illustrated on a wind turbine for tower load mitigation and rotor speed regulation. DS came up with the concept of modifying the CPC-based RDAC controller with an IPC-based controller to extend its scope to include blade load mitigation. With supervision from DS, EK designed the IPC controller and extended this concept with the idea from DS and JGN of using LQG-based switching controllers, to develop a novel aIPC controller for blade load mitigation. EK evaluated the proposed control scheme (RDAC+aIPC) on the 1.5 MW NREL reference wind turbine by running simulations and analyzing the obtained results. With valuable input from DS, EK wrote the manuscript. All authors provided important input to this work from concept to manuscript stage.



*Competing interests.* The authors declare that they have no conflict of interest.

*Acknowledgements.* This work is partly supported through a scholarship awarded to the first author by the German Academic Exchange Service (DAAD) in cooperation with the Ministry of Education of Kenya, for his Ph.D. study at the Chair of Dynamics and Control, UDE, Germany.



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
