# Peer review of "Adaptive robust observer-based control for structural load mitigation of wind turbines"

_Wind Energy Science, 2021_

## Referee Comment (RC1)

**Review of "Adaptive robust observer-based control for structural load mitigation of wind turbines" by Edwin Kipchirchir et al., manuscript number: wes-2021-143**

**General comments**

The manuscript presents a robust disturbance accommodation (RDAC) pitch control method for the above-rated operating region. The method is based on a previous publication by the authors with the addition of a decoupled 1P cyclic IPC loop. The theoretical basis of the controller design is explained. The, mainly qualitative, results include a wind-step simulation and one turbulent simulation of 200s comparing RDAC with RDAC+IPC in time and frequency domain as well as a covariance analysis of power with blade flapwise and tower fore-aft structural loads.

The scope of the study is not clear to me, as there is not a consistent link between motivation/hypothesis-methodology-conclusions. The RDAC is already presented in a previous publication from the authors while the decoupled IPC loop is rather standard with the addition of a Kalman filter to account for noise in state estimation. I think the scope and novelty have to be clarified and explained further.

The results are compared between the two new implementations only, without comparing with a conventional PI controller. This makes it hard to realize what the possible benefit would be for the wind energy community compared to the current status. Moreover, the simulation results are very limited including one step-wind simulation and one turbulent simulation of 200s, which in my opinion are not enough to evaluate the potential benefits.

The literature review is also limited, mostly using self-citations, and seems to be missing a large part of literature working on adaptive control design, DAC and IPC. Furthermore, the choice of the wind turbine model is not justified. Such a control scheme would probably be more relevant in larger turbines with larger and more flexible blades. The size of the machine and the related aeroelastic properties are not relevant for modern commercial systems (onshore or offshore). If the scope should be limited to onshore turbines (the choice has to be justified though) the IEA 3.4 MW or the NREL 5 MW r.w.t. could be used. Otherwise, the DTU 10MW or IEA 10MW/15MW machines can be also considered.

In my opinion, this work needs a thorough revision/rewriting to be accepted for publication. A more in-depth literature review, clarification on the scope, clear differentiation with previous work from the authors, methodology (more detailed explanations of the different implementations, reporting of values used, etc), and possibly a different WT model are some of the topics that need to be addressed. Nevertheless, my major concern is about section 5 which is not convincing. More simulations are required covering more operating conditions along with more relevant quantitative metrics for comparison. Moreover, the results should be compared with a tuned conventional PI pitch controller, as was also stated in the authors' previous work on the same topic. Finally, there are some minor issues with the terminology and phrasing used throughout the manuscript which I believe should be addressed in a later stage and are not discussed here.

**Specific comments**

Methodology

- Explain why the chosen WT model is relevant. My recommendation would be to switch to one of the most relevant in terms of turbine size and capacity (see previous comments).
- Be more specific in the description of the models and simulations: which FAST version is used, which DOFs are enabled and why etc.
- L 49-51 The sentence is not clear, seems like the wind turbine model and the aeroelastic software are mixed. Also, the meaning of "domicile" in this context is not clear.
- The pitch actuator dynamics are modeled as a first-order low pass filter, what is the time constant used? This choice is important to be stated and explained. In the current version, no value is discussed. In l150 the actuator modeling is referred again as a transfer function included in the plant. Are these the same, can you clarify? In general, provide specific values for constants and derived variables throughout section 3.
- How are the controllers and the switching implemented for both RDAC and IPC? Traditionally this is done based on the collective pitch angle. More explanations are needed to understand the method and ensure reproducibility.
- How is the switching between regions 2.5 and 3 implemented with the proposed RDAC?
- More explanation on the implementation of the method (switching, parameter choice, obtained values etc) are needed and the specific values applied should be provided along with the justification/derivation. In the current state, mainly symbolic derivations are included in the manuscript.
- In l160-162 the authors mention that the RDAC approach suggested is valid for a very narrow operational envelope. How is the smooth transition between the controllers impoemented? Can it be implemented in practice? How does it compare with the common gain scheduled PI CPC controller?
- Tower base fore-aft bending moment is not a standard measurement existing on every turbine. I understand that you used this to improve the model performance, but I think it should be at least mentioned. Did you try to use some of the already existing measurements or an observer instead?
- L 110-113 Possible methods to derive the gains are mentioned but it is not clear to me what methods were used in this work. Please be specific on what is used in this study and why. The text in section 3 reads more like a controls textbook rather than a specific application.
- Maybe I am missing something, but sections 3.1 and 3.2 seem to be the derivation of the RDAC similar to the previous publication from the authors (Do and Söffker, 2021) using also the exact same figure. Does this generic theory need to be repeated to its whole? It is not clear to me if the scope is the RDAC or the cyclic IPC in section 4. Please clarify the differences between the previous publication and state the novelty of the present work.
- Also in section 4, it is not clear to me how the distinct IPC controllers are combined. The sentence in l206-207 is not clear on this. Additionally, how is the incoming wind speed defined and measured? It could make sense to look into using the CPC value as an indicator to switch as it is common practice.
- How are stability and robustness guaranteed when the two methods (RDAC and IPC) are combined?

Results

- The manuscript refers in the introduction and abstract to load mitigation but in the results, no DELs are shown, and not enough arguments are made for the performance of the controller quantitatively. I would suggest focusing on DEL analysis following the IEC recommendations (in terms of wind speeds, TI, shear, duration, etc.) to quantify the possible benefits compared to the baseline.
- The analysis with the step wind is not serving the intended purpose. I don't see the purpose of comparing the PSDs or the time series of speed and power with the steps. Why would the power/speed be changing due to the IPC? How is robustness verified with the step simulations?
- The purpose of the power-load covariance analysis is not clear to me. The relevant figures (8 and 13) are difficult to read and hardly discussed in the manuscript. My suggestion is to remove this part or explain clearly its purpose.
- One turbulent simulation of 200s (including the initial transients) is not enough to show the effectiveness of the controller. More wind speeds and seeds have to be evaluated (see 1$^{st}$ comment of results)
- Specific information on the simulations like windfield generation method (Mann, Veers, etc.), dimensions and duration, DOFs and models activated in FAST, etc. have to be reported.
- The purpose of figures 9-11 is not clear to me. The IPC actuation can be evaluated with other metrics like actuator duty cycle, pitch angle standard deviation, pitch rate, total pitch travel, etc. The possible load reduction can not be identified by visually examining the time series.
- As the authors state the mean values are the same and the standard deviation is reduced by 12%. This is not enough to support the load reduction claims. DELs should be calculated taking into account the load cycles using a rainflow algorithm in longer simulations. I suggest using more wind conditions including more seeds per operating point.
- The load reduction should be discussed compared to a conventional pitch controller and not only between RDAC and RDAC+IPC.
- More load channels have to be evaluated in blades, tower bottom, and tower top. More concrete metrics about rotor speed, power, and pitch activity have to be used to evaluate quantitatively the effectiveness of the suggested methods with more simulations.
- Figure 11 shows overshoots of the power up to 25% and in general high fluctuations. Can this be considered good power/set point tracking? Again a comparison with the conventional controller could tell more about the quality of the proposed methods.
- The single turbulent simulation is only 200s long including the initial transients. I believe it is not enough and longer simulations are required to have meaningful PSD analyses and to derive metrics like DELs, standard deviations, actuator duty cycle, etc.
- Figure 13 is discussed in one sentence in L 286. Can you clarify what is its purpose and why it proves that the proposed controller improves structural load mitigation?

---

## Referee Comment (RC2)

**Review - Adaptive robust observer-based control for structural load mitigation of wind turbines**

Torben Knudsen, Aalborg University

April 21, 2022

**Summary**

The objectives of this paper is to improve on mitigating structural loading in rotor blades and tower with a good rotor speed and power regulation performance in the presence of model uncertainties and changing operating conditions. The main problem with the paper is that the improvement is demonstrated by comparing one already developed controller with a extension where both are made by the authors. Preferable it should be compared to results by other and or controllers made by others. On top of this the assumed known inputs as the (precise) hub wind speed and the tower based bending moment is not realistic. Also the assessment does not include the standard performance measures. The methods used in the paper are well know. Based on this and my comments below I at least suggest a major revision.

**Specific comments**

1. Abstract: "With growth in the physical size of wind turbines, an increased structural loading of wind turbine components affecting operational reliability is expected" Why is a small 1.5MW turbine used for testing instead of a more modern one e.g. the 10MW or 15MW IEA(DTU) RWT?

2. 3 Robust observer-based control:

   (a) The linearization is performed numerically by FAST. The FAST model has 16 DOF's. Only a subset is chosen for the control design model. Please motivate the choice of DOF's an corresponding states in (2).

   (b) When only the flap wise blade movement is included how is the IPC effect on the drive train modeled?

(c) There seems to be only one input namely collective pitch. However, besides blade pitch angles generator torque is also a control handle. This is often used to control drive train oscillations. Why is this not included?

(d) "The measurements y include rotor speed w and tower-base fore-aft bending moment." The tower bending moment is not a available measurement on commercial turbines but normally nacelle acceleration is. This means the setup is unrealistic?

(e) "Because pitch actuator dynamics are faster than other wind turbine dynamics, it is modeled as a first-order lag (PT1)" The pitch actuator is modeled as a first order low pas (LP) filter. That's fine but the most important part of the pitch actuator is normally not the time constant but the limited pitch rate of 5-15 deg/s?

3. 3.1 Disturbance accommodating control for wind turbines:

   (a) F= 0 in (7) means the disturbance is constant!
       i. How does this fit with the mentioned step?
       ii. What is the interpretation related to the real turbine physics?

4. 3.2 Robust disturbance accommodating control:

   (a) In figure 1 there is a known disturbance "Hub-height wind disturbance d". On commercial turbines the wind speed is only measured by the nacelle anemometer which are very uncertain mainly do to being just behind the rotor. Please explain how this is accounted fore?

5. 4.1 Adaptive independent pitch control:

   (a) "As wind turbine rotor blades rotate, they experience varying aerodynamic loads at different azimuth positions due to vertical wind shear" The spatial variations will be slowly time varying. Maybe the vertical wind shear is the main effect depending on the site. In a wind farm, where most turbines are located, horizontal shear do to partial wakes might be as important as vertical shear. Please motivate the focus here?

6. 5.1 Performance measures for analyzing results:

   (a) The standard measure for fatigue loading is damage equivalent load (DEL) calculated using rain flow counting (RFC). Please explain why this measure is not even mentioned?

   (b) Please also include the actuator activity e.g. measured with total traveled pitch angles.

   (c) Drive train loads should also be evaluated?

7. 5.1.2 Frequency domain:

(a) This section seems to explain the Welch method even though there is a reference. Is this necessary?

8. 5.2 Step wind profile results:

(a) Wind speed steps are not realistic! Please explain the value of this?

---

## Author Comment (AC1)

|  | Reviewers' comments | Reply to the editor |
|---|---|---|
| Anonymous Referee 1 | 1. The manuscript presents a robust disturbance accommodation (RDAC) pitch control method for the above-rated operating region. The method is based on a previous publication by the authors with the addition of a decoupled 1P cyclic IPC loop. The theoretical basis of the controller design is explained. The, mainly qualitative, results include a wind-step simulation and one turbulent simulation of 200s comparing RDAC with RDAC+IPC in time and frequency domain as well as a covariance analysis of power with blade flapwise and tower fore-aft structural loads. | We thank the reviewer for suitable summary of our contribution. |
|  | 2. The literature review is also limited, mostly using self-citations, and seems to be missing a large part of literature working on adaptive control design, DAC, and IPC. Furthermore, the choice of the wind turbine model is not justified. Such a control scheme would probably be more relevant in larger | As usual to avoid self-plagiarism and to clearly differentiate this contribution from previous one, own references must be given. The self-citation rate is below 20 %. We believe this should be acceptable.

Further: The literature review will be improved by incorporating references from suggested areas. |

| | | |
|---|---|---|
| | turbines with larger and more flexible blades. The size of the machine and the related aeroelastic properties are not relevant for modern commercial systems (onshore or offshore). If the scope should be limited to onshore turbines (the choice has to be justified though) the IEA 3.4 MW or the NREL 5 MW r.w.t. could be used. Otherwise, the DTU 10MW or IEA 10MW/15MW machines can be also considered | In this contribution, the 1.5 MW wind turbine model is chosen as it meets the threshold in power rating for what can be considered a commercial wind turbine. Although its size does not correspond to the current state-of-the art in onshore wind, the control strategy proposed can be applied in controlling larger wind turbines. Therefore the wind turbine model serves as an example. The NREL 5 MW RWT will be considered in future work. |
| | 3. In my opinion, this work needs a thorough revision/rewriting to be accepted for publication. A more in-depth literature review, clarification on the scope, clear differentiation with previous work from the authors, methodology (more detailed explanations of the different implementations, reporting of values used, etc), and possibly a different WT model are some of the topics that need to be addressed. Nevertheless, my major concern is about section 5 which is not convincing. More simulations are required covering more operating conditions along with more relevant quantitative metrics for comparison. Moreover, the results should be compared with a tuned conventional PI pitch | The scope of this contribution including a clear demarcation with previous work will be clarified both in the abstract and the introduction section.

In contradiction to the previous statement of the reviewer, here the self-referencing is ok. This is fine. We will point out more clearly the new points of this contribution. More details on the controller implementation will be provided.

Regarding section 5, improvements can be made. Results obtained from simulations ran using more wind field realizations will be discussed. As suggested, relevant quantitative evaluation metrics like DELs and actuator duty cycle will be included.

Comparison between the proposed controller and the gain-scheduled PI-based NREL 1.5 MW CPC baseline controller will be done. In the current contribution RDAC is compared with the proposed control scheme RDAC+aIPC.

Although a smaller wind turbine has been considered, the proposed control strategy can be used to control larger turbines with similar |

| | | |
|---|---|---|
| | controller, as was also stated in the authors' previous work on the same topic. Finally, there are some minor issues with the terminology and phrasing used throughout the manuscript which I believe should be addressed in a later stage and are not discussed here. | configuration. Future work will be based on the larger NREL 5 MW RWT. |
| Methodology | Specific objectives

Explain why the chosen WT model is relevant. My recommendation would be to switch to one of the most relevant in terms of turbine size and capacity (see previous comments). | The 1.5 MW wind turbine model is chosen in this contribution as it meets the threshold for what can be considered a commercial wind turbine. Although its size does not correspond to the current state-of-the art in onshore wind, the proposed control strategy can be applied controlling larger wind turbines. The NREL 5 MW RWT will be considered in future work. |
| | Be more specific in the description of the models and simulations: which FAST version is used, which DOFs are enabled and why etc. | The FAST version used (version 7) is stated in L 50-51.

The assumption made was that by simply stating the states included in the linear model (Eq. 2) one would directly know which DOFs are enabled. The reasons are provided in L 69-71. It is to capture the most important dynamics related to load mitigation in wind turbine blades and tower as well as generator speed regulation, while simulating a flexible drivetrain. This is done while avoiding unnecessary complexity in the linear model (3). For clarity, additional statements will be included in the manuscript. |
| | L 49-51 The sentence is not clear, seems like the wind turbine model and the aeroelastic software are mixed. Also, the meaning of "domicile" in this context is not clear. | This sentence will be rephrased to give more clarity. The word "domicile" is originally used in one of the FAST documentations to mean "included or available". A more suitable word can be used. |

| | | |
|---|---|---|
| | The pitch actuator dynamics are modeled as a first-order low pass filter, what is the time constant used? This choice is important to be stated and explained. In the current version, no value is discussed.

In L 150 the actuator modeling is referred again as a transfer function included in the plant. Are these the same, can you clarify? In general, provide specific values for constants and derived variables throughout section 3. | The actuator, which is modeled as a $1^{st}$ order low-pass-filter has a time constant of 0.2 seconds to simulate the slow dynamics of the pitch actuator compared to other wind turbine dynamics. Although the extension of the linear model has been considered in previous works, in this work, it is included as a transfer function to the generalized plant P, as stated in L 150. Therefore, extension of the model (Eq. 3) with the actuator model is erroneous and will be rectified. |
| | How are the controllers and the switching implemented for both RDAC and IPC? Traditionally this is done based on the collective pitch angle. More explanations are needed to understand the method and ensure reproducibility | Switching between the different IPC controllers is realized in Simulink using if-else logic. Depending on the prevailing wind speed measurement, the relevant controller, which is effective over a specified wind speed bin (see Table 2) is activated. The wind speed bins are only used for thresholding, hence typically inaccurate anemometer measurement should suffice. Switching between the 5 IPC controllers based on the incoming hub-height stochastic wind speed constitutes aIPC. The relevant explanation can be found in L 169-173. For more clarity, additional statements will be included. |
| | How is the switching between regions 2.5 and 3 implemented with the proposed RDAC? | The scope of this contribution is rotor speed regulation and tower and blade load mitigation. Switching between regions 2.5 and 3 is not implemented. |
| | More explanation on the implementation of the method (switching, parameter choice, obtained values etc.) are needed | The methodology will be improved to include more details as suggested. A figure to illustrate the switching implementation will be added |

| | | |
|---|---|---|
| | and the specific values applied should be provided along with the justification/derivation. In the current state, mainly symbolic derivations are included in the manuscript | |
| | In L 160-162 the authors mention that the RDAC approach suggested is valid for a very narrow operational envelope. How is the smooth transition between the controllers implemented? Can it be implemented in practice? How does it compare with the common gain scheduled PI CPC controller? | The RDAC controller is designed for rotor speed regulation and tower load mitigation, achieved using a CPC signal. It provides the main control signal to meet these objectives. On the other hand, the switching-based aIPC is used to mitigate periodic loading in the blades due to wind shear. The IPC control signals from aIPC are added the CPC signal from RDAC. The overall control signals are used to independently manipulate each blade to achieve the desired objectives. Relevant explanation can be found in L 209-215. This can be implemented in practice using two control loops, Figure 3.

In this contribution, proposed RDAC+aIPC controller is evaluated against RDAC controller and shows improvement in both load mitigation and speed regulation. A comparison between the proposed controller and the gain-scheduled PI-based NREL 1.5 MW CPC controller will be included. |
| | Tower base fore-aft bending moment is not a standard measurement existing on every turbine. I understand that you used this to improve the model performance, but I think it should be at least mentioned. Did you try to use some of the already existing measurements or an observer instead? | It is true that tower-base fore-aft bending moment is not a standard wind turbine measurement. However, it is considered in this contribution to implement load mitigation since it fits into the state-space scheme and is required for the given task (load mitigation). Direct measurement of this load is utilized since it is an available measurement channel in FAST. However, to implement full-state feedback, observers are designed for estimating all the states in both the RDAC and aIPC controllers.
In future work, nacelle accelerometer measurements will be considered. |

| | | |
|---|---|---|
| | L 110-113 Possible methods to derive the gains are mentioned but it is not clear to me what methods were used in this work. Please be specific on what is used in this study and why. The text in section 3 reads more like a controls textbook rather than a specific application. | The generalized plant P (Figure 1), which includes a connection of the wind turbine model, actuator dynamics, and weighting functions, is interconnected with the observer-based DAC model (Eq. 12), which carries the tunable elements (gains K, L) using lower Linear Fractional Transformation (LFT).  Non-smooth H-infinity optimization is then used to tune these gains by minimizing the maximum singular value of the transfer function from wind disturbance d to the controlled outputs z1, z2, z3.
This explanation will be added to the manuscript. |
| | Maybe I am missing something, but sections 3.1 and 3.2 seem to be the derivation of the RDAC similar to the previous publication from the authors (Do and Söffker, 2021) using also the exact same figure. Does this generic theory need to be repeated to its whole? It is not clear to me if the scope is the RDAC or the cyclic IPC in section 4. Please clarify the differences between the previous publication and state the novelty of the present work. | In L 65, the brief repetition of the content in 3.1 and 3.2 is stated as being necessary for principal understanding of RDAC control design and therefore also referenced.
The scope of this contribution is to augment the previous RDAC controller designed for rotor speed regulation and tower load mitigation, with an aIPC controller, which is adaptive to changes in operating point, for blade load mitigation. The novelty of aIPC controller is that switching between a bank of IPC controllers is achieved based on prevailing wind condition. |
| | Also in section 4, it is not clear to me how the distinct IPC controllers are combined. The sentence in l206-207 is not clear on this. Additionally, how is the incoming wind speed defined and measured? It could make sense to look into using the CPC value as an indicator to switch as it is common practice. | The relevant explanation for switching is provided earlier in L 169-173.
Switching between the different IPC controllers is realized in Simulink using if-else logic. Depending on the prevailing wind speed measurement the relevant controller, which is effective over a given wind speed bin (see Table 2) is activated.
Wind speed is an available measurement in FAST. This would be nacelle anemometer measurements in real wind turbine. Although these measurements are highly uncertain, strict accuracy is not required for switching |

| | | |
|---|---|---|
| | | |
| | How are stability and robustness guaranteed when the two methods (RDAC and IPC) are combined? | Both RDAC and aIPC controller gains are designed using control methods that guarantee closed-loop stability. Non-smooth H-infinity synthesis method is used for designing an optimal RDAC controller that is robust against model uncertainties and nonlinearities. To ensure performance in the entire region 3 operation, each of the 5 IPC controllers are designed using linear models extracted from operating points defined by wind speeds that cover above-rated operation (see Table 2). Wind speed bins define the thresholds for switching between IPC controllers.

Therefore, the combined closed loop stability is guaranteed since closed loop stability in each controller is guaranteed. |
| | The manuscript refers in the introduction and abstract to load mitigation but, in the results, no DELs are shown, and not enough arguments are made for the performance of the controller quantitatively. I would suggest focusing on DEL analysis following the IEC recommendations (in terms of wind speeds, TI, shear, duration, etc.) to quantify the possible benefits compared to the baseline. | As suggested, analysis using DELs will be included. |
| | The analysis with the step wind is not serving the intended purpose. I don't see the purpose of comparing the PSDs or the time series of speed and power with the steps. | Step wind simulation is used to evaluate transient performance. Step inputs are one of the usual inputs used in control for system analysis and suitable excitation.
PSD evaluation is relevant for evaluating the control performance at1P and 3P frequencies, which are the dominant frequencies that |

| | | |
|---|---|---|
| | Why would the power/speed be changing due to the IPC? How is robustness verified with the step simulations? | excite vibrations in the blades and tower, respectively. Lower magnitudes at these frequencies indicates reduction in structural loading. It is true that step wind is not suitable for evaluating robustness due to its stationarity, which is not the only purpose here. |
| | The purpose of the power-load covariance analysis is not clear to me. The relevant figures (8 and 13) are difficult to read and hardly discussed in the manuscript. My suggestion is to remove this part or clearly explain its purpose. | Section 5.1.3 discusses the power-load covariance performance metric for evaluating the effectiveness of the proposed control strategy. First proposed in [Do and Söffker 2020], the method evaluates the overall performance and relationship in power regulation and load mitigation. More details on this can be found in the cited reference.

We do not intend to remove this, because this clearly illustrates the differences of different controllers for real inputs. |
| | One turbulent simulation of 200s (including the initial transients) is not enough to show the effectiveness of the controller. More wind speeds and seeds have to be evaluated (see 1st comment of results). | Additional, longer duration wind speeds and random seeds will be used to evaluate the performance of the proposed control scheme. |
| | Specific information on the simulations like wind field generation method (Mann, Veers, etc.), dimensions and duration, DOFs and models activated in FAST, etc. have to be reported. | The wind turbine model used and DOFs activated during simulation will be provided. Details of the stochastic wind field used are given in L 270-273. Statistical details of the wind field will be added. |
| | The purpose of figures 9-11 is not clear to me. The IPC actuation can be evaluated with other metrics like actuator duty cycle, pitch angle standard deviation, pitch rate, total | As suggested additional quantitative metrics for evaluating speed regulation and load reduction especially for stochastic simulation will be additively considered. Load reduction has been evaluated using standard deviation. DEL analysis will be included. |

| | | |
|---|---|---|
| | pitch travel, etc. The possible load reduction cannot be identified by visually examining the time series. | |
| | As the authors state the mean values are the same and the standard deviation is reduced by 12%. This is not enough to support the load reduction claims. DELs should be calculated taking into account the load cycles using a rain-flow algorithm in longer simulations. I suggest using more wind conditions including more seeds per operating point. | Damage Equivalent Loads analysis as well as use of more wind field realizations will be additively considered. |
| | The load reduction should be discussed compared to a conventional pitch controller and not only between RDAC and RDAC+IPC | In the current contribution RDAC is compared with the proposed control scheme RDAC+aIPC. A comparison between the proposed control strategy and the 1.5 MW NREL baseline controller will be given. |
| | More load channels have to be evaluated in blades, tower bottom, and tower top. More concrete metrics about rotor speed, power, and pitch activity have to be used to evaluate quantitatively the effectiveness of the suggested methods with more simulations. | Additional load channels in blades and tower will be included in evaluating the effectiveness of the proposed control strategy. Additionally, suggested methods for evaluating speed/power regulation and pitch activity will be considered. |
| | Figure 11 shows overshoots of the power up to 25% and in general high fluctuations. Can this be | Yes, indeed power overshoots as high as 25 % is unacceptable. A mistake was made in that the measurement channel used to come up with plot in Fig 11b is rotor power. Since the 1.5 MW NREL wind |

| | | |
|---|---|---|
| | considered good power/set point tracking? Again, a comparison with the conventional controller could tell more about the quality of the proposed methods. | turbine has a generator efficiency of 94.4 %, this can partly explain the overshoots. Additionally, it is important to note that a stochastic wind profile that simulates an extreme case of TI of 17 % (low occurrence probability) is used. This drives the dynamics of the wind turbine in some instances beyond cut-out wind speed (above 25 m/s close to 29 m/s, Figure 9 a). Since a shutdown event is not simulated, power is bound to shoot beyond tolerable limits. More realistic wind conditions will be considered. A comparison with baseline controller will also be included. |
| | The single turbulent simulation is only 200s long including the initial transients. I believe it is not enough and longer simulations are required to have meaningful PSD analyses and to derive metrics like DELs, standard deviations, actuator duty cycle, etc. | Wind field realizations with longer simulations will be used to conduct PSD and DEL analysis |
| | Figure 13 is discussed in one sentence in L 286. Can you clarify what is its purpose and why it proves that the proposed controller improves structural load mitigation? | The power-load covariance method proposed in [Do and Söffker 2020], evaluates the overall controller performance using ellipse iso-contours. This gives a clear illustration of the relationship between power regulation and load mitigation, which are the main objectives addressed by the proposed controllers. |
| Reviewer 2 Torben Knudsen, Aalborg | The objectives of this paper is to improve on mitigating structural loading in rotor blades and tower with a good rotor speed and power regulation performance in the presence of model uncertainties and changing operating conditions. The main problem with the paper is that the improvement is | Comparison between the proposed controller and the gain-scheduled PI-based NREL 1.5 MW CPC baseline controller will be done. As it is this contribution shows improvement over the previous control strategy RDAC, which has been compared with a DAC controller.

The assumption that hub-height wind speed is precisely known is not realistic, however, since this is only used realize switching between different controllers in aIPC controller, inaccurate |

| | |
|---|---|
| demonstrated by comparing one already developed controller with a extension where both are made by the authors. Preferable it should be compared to results by other and or controllers made by others. On top of this the assumed known inputs as the (precise) hub wind speed and the tower based bending moment is not realistic. Also the assessment does not include the standard performance measures. The methods used in the paper are well know. Based on this and my comments below I at least suggest a major revision. | anemometer measurements should suffice. Additionally, tower-base fore-aft bending moment measurement is not realistic but is used to demonstrate the concept. While tower fore-aft acceleration measurement is more practical, this does not fit into the state-space scheme.

While the individual proposed methods are well known, the combined control strategy achieving a wider scope of objectives has not been realized before. Additionally, the novelty of aIPC controller is that switching between a bank of IPC controllers is achieved based on prevailing wind condition. |
| **Abstract:**

"With growth in the physical size of wind turbines, an increased structural loading of wind turbine components affecting operational reliability is expected" Why is a small 1.5MW turbine used for testing instead of a more modern one e.g. the 10MW or 15MW IEA(DTU) RWT? | In this contribution, the 1.5 MW wind turbine model is chosen as it meets the threshold in power rating for what can be considered a commercial wind turbine. Although its size does not correspond to the current state-of-the art in onshore wind, the control strategy proposed can be applied in controlling larger wind turbines, which have a similar configuration. The NREL 5 MW RWT will is being for considered in future work. |
| **3 Robust observer-based control:**

The linearization is performed numerically by FAST. The FAST model has 16 DOF's. Only a subset | L69-70 highlights the reason for choosing 6 DOFs out of the available 16. This statement will be improved. It is to capture the most important dynamics related to load mitigation in wind turbine blades and tower as well as generator speed regulation, while simulating a flexible drivetrain. This is done while avoiding unnecessary complexity in the linear model (3). |

| | | |
|---|---|---|
| | is chosen for the control design model. Please motivate the choice of DOF's an corresponding states in (2). | |
| | When only the flap wise blade movement is included how is the IPC effect on the drive train modeled? | The effect of tower side-side motion on drive-train torsion is well known. In this work, the effect of IPC on the drivetrain is not studied, this can be evaluated during simulation by enabling the tower side-side DOF. |
| | There seems to be only one input namely collective pitch. However, besides blade pitch angles generator torque is also a control handle. This is often used to control drive train oscillations. Why is this not included? | While generator torque control method is mainly used for limiting drivetrain loads in below-rated wind speed regime, the proposed control strategy is only applied to limit tower and blade loads and regulate rotor speed in region III. Here, the generator torque is held constant. |
| | "The measurements y include rotor speed w and tower-base fore-aft bending moment." The tower bending moment is not an available measurement on commercial turbines but normally nacelle acceleration is. This means the setup is unrealistic? | Indeed, it is not practical to rely on tower bending moment measurement especially in large wind turbines. This load channel is used for convenience since it fits the state-space scheme. It is also required for the task of tower load mitigation. However, relying on tower-top displacement information from nacelle accelerometer is expected to provide the same insight into the influence of wind field on tower frequency response
In future work, nacelle accelerometer measurements will be considered. |
| | "Because pitch actuator dynamics are faster than other wind turbine dynamics, it is modeled as a first-order lag (PT1)" The pitch actuator is modeled as a first order low pas (LP) filter. That's one but the most important part of the pitch actuator | Thank you for pointing out this. Information on whether the proposed method violates the turbine actuator pitch-rate limit will be included |

| | | |
|---|---|---|
| | is normally not the time constant but the limited pitch rate of 5-15 deg/s? | |
| | **3.1 Disturbance accommodating control for wind turbines:**

F= 0 in (7) means the disturbance is constant!

i. How does this fit with the mentioned step?

ii. What is the interpretation related to the real turbine physics? | Although wind disturbance model used is simple, the constant model in combination with high gains is the most flexible model able to estimate unknown disturbances with unknown dynamics like uniform changes in hub-height wind speed. Since this model is augmented to the CPC-based linear model (3), only the collective wind component at the hub-height is considered. |
| | **3.2 Robust disturbance accommodating control:**

 In figure 1 there is a known disturbance "Hub-height wind disturbance d". On commercial turbines the wind speed is only measured by the nacelle anemometer which are very uncertain mainly due to being just behind the rotor. Please explain how this is accounted for? | Controller design is based on a linear model of the wind turbine, in which a steady wind of 18 m/s defines the operating point for extracting this model (L67-69). Since a robust controller (RDAC) is being considered, it is expected that the closed-loop system with the designed robust controller should perform well in an unknown wind field around the working point. Therefore, for simulation the wind disturbance d (generated using TurbSim) is unknown to the RDAC controller.
For the aIPC controller, switching between different IPC controllers is achieved based on hub-height wind speed measurement. However, since wind speed bins (Table 2, column 2) are used for thresholding, hence accurate measurements are not needed, anemometer measurements (with limited accuracy) should suffice in real applications.
This clarity will be provided in the document. |
| | **4.1 Adaptive independent pitch control:** | In this contribution, vertical wind shear is only considered since turbine level control is implemented. However, in a wind farm scenario, wind wakes can cause horizontal shear which contributes |

| | | |
|---|---|---|
| | "As wind turbine rotor blades rotate, they experience varying aerodynamic loads at different azimuth positions due to vertical wind shear" The spatial variations will be slowly time varying. Maybe the vertical wind shear is the main effect depending on the site. In a wind farm, where most turbines are located, horizontal shear due to partial wakes might be as important as vertical shear. Please motivate the focus here? | to periodic loading. This is neglected in this work since this cannot be simulated in the standalone 1.5 MW wind turbine. However, the concept implemented in the aIPC controller for handling periodic loads due to wind shear or veer would work in both cases. |
| | **5.1 Performance measures for analyzing results:**

The standard measure for fatigue loading is damage equivalent load (DEL) calculated using rain flow counting (RFC). Please explain why this measure is not even mentioned? | Thank you for pointing on this. DEL evaluation will be additionally included as part of the performance evaluation of the proposed method |
| | Please also include the actuator activity e.g., measured with total traveled pitch angles. | Thank you for this addition. Pitch actuator activity will be evaluated |
| | Drive train loads should also be evaluated? | Although mitigating drive-train loads is not considered in this contribution, influence of the proposed method on this component will be evaluated |
| | **5.1.2 Frequency domain:** | Welch's method for performing PSD analysis is briefly explained for principal understanding. However, details are found in the cited reference [Welch, 1967]. |

| | | |
|---|---|---|
| | This section seems to explain the Welch method even though there is a reference. Is this necessary? | |
| | **5.2 Step wind profile results:**

Wind speed steps are not realistic! Please explain the value of this? | A step wind profile is only used to evaluate transient performance of the proposed method.
Step inputs are one of the usual inputs used in control for system analysis and suitable excitation.
However, a stochastic wind profile provides a more realistic evaluation of the proposed method. |